# Effects of Topical Anaesthetic and Buccal Meloxicam Treatments on Concurrent Castration and Dehorning of Beef Calves

**DOI:** 10.3390/ani8030035

**Published:** 2018-02-28

**Authors:** Dominique Van der Saag, Peter White, Lachlan Ingram, Jaime Manning, Peter Windsor, Peter Thomson, Sabrina Lomax

**Affiliations:** 1Sydney School of Veterinary Science, Faculty of Science, The University of Sydney, Sydney, NSW 2006, Australia; p.white@sydney.edu.au (P.Wh.); peter.windsor@sydney.edu.au (P.Wi.); 2School of Life and Environmental Sciences, Faculty of Science, The University of Sydney, Sydney, NSW 2006, Australia; lachlan.ingram@sydney.edu.au (L.I.); jaime.manning@sydney.edu.au (J.M.); peter.thomson@sydney.edu.au (P.T.); sabrina.lomax@sydney.edu.au (S.L.)

**Keywords:** behaviour, castration, cattle, dehorning, buccal meloxicam, pain, topical anaesthetic, weight gain

## Abstract

**Simple Summary:**

The pain caused by surgical procedures performed routinely for managing livestock husbandry is recognised as a significant animal welfare issue for food security. In recent years, there has been progress encouraging the uptake of pain relief in extensively managed livestock operations, with research and development offering options for the practical delivery of anaesthetics and analgesics during these procedures. In Australia, topical anaesthetic and buccal meloxicam treatments are now commercially available for use during routine surgical husbandry procedures of lambs and calves. A study to assess the effect of these treatments on weight gain and behavioural variables following concurrent surgical castration and amputation dehorning in beef calves is reported. Results showed that a combination of topical anaesthetic and buccal meloxicam appeared to reduce pain following castration and dehorning, with improved weight gain and increased lying activity in the first few days following the procedures. In addition, some individual behaviours expressed by the calves on the day of treatment suggested pain was relieved by topical anaesthetic and buccal meloxicam, although further clarification of this observation is required. These findings demonstrate that provision of topical anaesthetic and buccal meloxicam to beef calves undergoing surgical castration and amputation dehorning can result in improved animal welfare and production.

**Abstract:**

The use of pain relief during castration and dehorning of calves on commercial beef operations can be limited by constraints associated with the delivery of analgesic agents. As topical anaesthetic (TA) and buccal meloxicam (MEL) are now available in Australia, offering practical analgesic treatments for concurrent castration and dehorning of beef calves, a study was conducted to determine their efficacy in providing pain relief when applied separately or in combination. Weaner calves were randomly allocated to; (1) no castration and dehorning/positive control (CONP); (2) castration and dehorning/negative control (CONN); (3) castration and dehorning with buccal meloxicam (BM); (4) castration and dehorning with topical anaesthetic (TA); and (5) castration and dehorning with buccal meloxicam and topical anaesthetic (BMTA). Weight gain, paddock utilisation, lying activity and individual behaviours following treatment were measured. CONP and BMTA calves had significantly greater weight gain than CONN calves (*p* < 0.001). CONN calves spent less time lying compared to BMTA calves on all days (*p* < 0.001). All dehorned and castrated calves spent more time walking (*p* = 0.024) and less time eating (*p* < 0.001) compared to CONP calves. There was a trend for CONP calves to spend the most time standing and CONN calves to spend the least time standing (*p* = 0.059). There were also trends for the frequency of head turns to be lowest in CONP and BMTA calves (*p* = 0.098) and tail flicks to be highest in CONN and BM calves (*p* = 0.061). The findings of this study suggest that TA and MEL can potentially improve welfare and production of calves following surgical castration and amputation dehorning.

## 1. Introduction

Dehorning and castration are routine husbandry procedures that are performed in the global cattle industries. These procedures are of particular importance to northern Australian beef properties, where *Bos indicus* breeds are dominant and herds are managed on extensive areas of land [1]. Dehorning is still a necessary procedure in northern Australia where there are low numbers of polled animals due to the complex mode of inheritance of the poll gene in *Bos indicus* breeds [2]. Castration is particularly important on northern Australian beef properties as the extensive nature of farming practices makes separation of males and females unfeasible [1]. On these properties, it is common for calves to be mustered only once or twice a year for weaning and ‘marking,’ the latter procedure involving ear tagging, ear notching, branding, dehorning and castrating [1]. The infrequency of mustering results in large numbers of calves being processed rapidly for marking procedures., with variation in ages from a few months and up to 10 months old [1]. Pain associated with the marking procedures, particularly castration and dehorning, is considered a significant welfare issue for the Australian beef industry. It is particularly concerning when routine marking involves older calves where the testicular tissues and horns are more developed than younger calves [1]. Although injectable anaesthetics and analgesics may provide pain relief for these procedures, this approach is not considered a practical option on northern Australian beef properties [1]. 

The need for provision of a practical method of delivering pain relief in livestock systems has been recognised for over a decade in Australia. Following extensive research, the topical anaesthetic gel Tri-Solfen^®^ (Bayer Animal Health, Pymble, NSW, Australia), containing lignocaine and bupivacaine, is now commercially available for use during various livestock husbandry procedures, including application to mulesing and tail docking wounds in lambs [3,4,5] and for surgical castration wounds in both lambs and calves [3,6,7,8]. Similarly, for practical reasons, a buccal gel formulation of the non-steroidal anti-inflammatory drug (NSAID) meloxicam, Ilium^®^ Buccalgesic OTM (Troy Laboratories, Glendenning, NSW, Australia), is registered for use during surgical castration of lambs [9] and calves and tail docking of lambs [9]. The topical anaesthetic (TA) has previously reduced post-operative wound sensitivity for at least 24 h and pain-related behaviours for at least 4 h following surgical castration in beef calves [6]. The efficacy of TA in desensitising beef calf dehorning wounds for 6 h following the procedure has been suggested as comparable to a cornual nerve block of lignocaine [10]. The TA and the buccal meloxicam (MEL), administered separately and in combination, have been shown to reduce some pain-related behaviours during a 5-h period following surgical castration in beef calves [8]. The MEL has also been shown to reduce maximum scrotal temperature 2 days following surgical castration, suggested due to an anti-inflammatory effect [8]. The TA is applied during or immediately following the procedures using a spray applicator and is absorbed across open wounds and mucosal tissue. The buccal meloxicam (MEL) is administered before or during the procedures using a gun applicator and is absorbed through the oral mucosa. The use of these products only adds a number of seconds onto processing each animal. Therefore, both methods of anaesthetic and analgesic delivery, result in an easier, faster and safer drug administration process compared to pre-surgical administration of injections. 

This study aimed to assess the effects of TA and MEL, separately and in combination, on weight gain and behavioural variables following concurrent surgical castration and amputation dehorning of *Bos indicus* weaner calves in an extensively managed system. We hypothesised that TA and BM would improve weight gain and reduce pain-related behaviours following concurrent surgical castration and amputation dehorning, especially when administered in combination. 

## 2. Materials and Methods

The experimental protocol was approved by the Animal Ethics Committee of the University of Sydney (Approval No. 5832) and was conducted in accordance with the guidelines of the ‘Australian code for the care and use of animals for scientific purposes’ [11]. Two experiments were conducted using *Bos indicus* or *Bos indicus* crossbred weaner bull calves (approximately 6–8 months of age). All animals were sourced from a commercial beef herd in QLD, Australia and were undergoing routine weaning and marking (as previously described). One week prior to commencement of experiment 1, all calves were mustered, separated from their mothers and held in a set of ‘weaning yards’ with *ad libitum* access to water and lucerne hay, as is commonly practiced on northern Australian beef herds.

### 2.1. Treatments and Experimental Design

For both experiments, calves were randomly allocated to one of five treatments in the order that they were processed by use of random numbers generated in Microsoft Excel 2007 (Microsoft Corporation, Redmond, WA, USA): (1) no castration or dehorning/positive control (CONP); (2) castration and dehorning/negative control (CONN); (3) castration and dehorning with pre-operative buccal meloxicam (BM); (4) castration and dehorning with intra-operative topical anaesthetic (TA); and (5) castration and dehorning with pre-operative buccal meloxicam and intra-operative topical anaesthetic (BMTA). There were 50 calves per treatment group for experiment 1. A subset of these calves (20 per treatment group) was fitted with global positioning system (GPS) units and a further subset of these calves (10 per treatment group) was fitted with accelerometers. In experiment 2, there were 12 calves in the CONP and BMTA treatment groups and 11 calves in the CONN, BM and TA treatment groups.

Experiment 1 was performed over 7 days, from the day of treatment (day 0) to 6 days post-treatment (day 6). On day 0, calves were processed through a race where they were weighed using cattle scales, Livestock Manager TSi 2 (Gallagher Group Ltd., Hamilton, New Zealand) within the cattle crush, Ultimate Crush (RPM Australia-Pacific Pty Ltd., Gatton, QLD, Australia). They were restrained in a head bale for ear tagging and ear notching. BM and BMTA calves were treated with buccal meloxicam (MEL) at this point. Calves were then moved through a separate race to a weaner cradle (Morrissey & Co. Calves Handling Equipment, Jandowae, QLD, Australia) where they were restrained in left lateral recumbency for treatment and attachment of GPS and accelerometer units. 

Commercially produced CatLog^TM^ GPS units (17 × 25 × 5 mm) (Catnip Technologies Ltd., Anderson, CA, USA), designed for use on domestic cats and their attached battery packs (17 × 20 × 49 mm), were placed in acrylonitrile butadiene styrene plastic Jiffy enclosure boxes (130 × 68 × 44 mm), Jiffy box (Jaycar Electronics, Rydalmere, NSW, Australia) and secured in place with Styrofoam. The boxes were then enclosed with their supplied lids and fixing screws and attached to Gripwell luggage straps (25 mm × 2 m) (Gripwell Australia Pty Ltd., Chatswood, NSW, Australia), using zip ties. On day 0, a single luggage strap was secured around the neck of each animal with the plastic box positioned on the upper right side of the neck to ensure the GPS antenna was unobstructed from satellite signals. In addition, the plastic boxes were fixed in place on the neck of the animal with Parfix^®^ fast grip contact adhesive (DeluxGroup Ltd., Clayton, VIC, Australia). On day 6, the luggage straps and boxes were quickly and carefully removed from the cattle by cutting the straps using a knife. These GPS units were used as they are lightweight (22 g) and low-cost and therefore practical and cost-effective for tracking greater numbers of individual animals [12].

HOBO Pendant G Acceleration Data Loggers (Onset Computer Corporation, Bourne, MA, USA), were inserted into pieces of foam sponge and secured on each calf to the lateral aspect of the right hind leg proximal to the fetlock using 3M™ VetRap™ Bandaging Tape (Medshop Australia, Preston, VIC, Australia) and Norton Bear 50 mm × 15 m Silver Cloth Tape (Saint Gobain, Somerton, VIC, Australia). The units were positioned such that the x-axis was perpendicular to the ground and pointing ventrally, the y-axis was parallel to the ground and pointing cranially and the z-axis was parallel to the ground and pointing toward the midplane. 

All calves except CONP calves were castrated and dehorned and TA and BMTA calves were treated with topical anaesthetic (TA), as described below, whilst still restrained in the cradle. Calves were released into another holding yard (300 m^2^) where they remained until the last animal was processed. 

The marking process commenced at 07:30:00 a.m. and was concluded by 05:00:00 p.m. hours. When all calves had been processed, they were moved into a laneway (700 m^2^) where they remained until 06:00:00 a.m. the following day when they were moved to a large paddock (619 ha) for a further 6 days. During this time, calves had *ad libitum* access to pasture and water. On day 6 at 06:00:00 a.m., calves were mustered back into the holding yards adjacent to the handling facilities and processed through the first race. Whilst in the race, GPS and accelerometer units were removed, then the calves were weighed in the cattle crush and released.

Experiment 2 was conducted over 3 days (days A, B and C), with 17 calves (3–4 per treatment group) treated on day A and 20 calves (4 per treatment group) treated on days B and C. Each day, calves were processed as per experiment 1. Calves were individually numbered on both sides and the back of the body with Dy-Mark Spray & Mark 350 g Spray Paint (Dy-Mark, Darra, QLD, Australia) while in the race. Following treatment, calves were released into a holding yard (104 m^2^) for behavioural recording, as described below. This process commenced at 07:30:00 a.m. and concluded by 08:30:00 a.m.

### 2.2. Castration and Dehorning

Castration and dehorning were performed by experienced technicians. Due to technician availability, the technicians were different for experiments 1 and 2. Castration was performed by pushing the testicles to the distal end of the scrotum and incising the scrotal skin and tunica dartos from the base and up each side with a scalpel blade and then the tunica vaginalis to expose the testes. Each testicle was then extruded through the openings to expose and sever the spermatic cords approximately 10 cm proximal to the head of the epididymis using the scalpel blade. Dehorning was performed using a Dominion Yearling Cup dehorner, (The Farm Store, Campbellfield, VIC, Australia). Dehorning was conducted by opening the cup, placing it over the horn, applying downward pressure and closing the handles to remove the horn tissue and immediate surrounding skin. The scalpel blade and the cup dehorner were chemically sterilised between use on each animal.

### 2.3. Analgesic Products

For the MEL, a gel formulation of meloxicam (10 mg/mL) as Ilium Buccalgesic^®^ (Troy Laboratories, Glendenning, NSW, Australia) was administered (0.5 mg/kg BW, rounded up to the nearest 50 kg BW) by a single experienced technician via a hook nozzle into the buccal pouch for absorption through the oral mucosa. Buccal meloxicam was administered 1 to 2 and 0.5 to 1 h prior to castration and dehorning, for experiments 1 and 2, respectively.

For the TA, a gel formulation containing lignocaine (40.6 g/L), bupivacaine (4.2 g/L), cetrimide (5 g/L) and adrenaline (24.8 mg/L) as Tri-Solfen^®^ (Bayer Animal Health, Pymble, NSW, Australia) was administered by a single, experienced technician via a spray applicator, where approximately 4 mL was applied for castration and another 4 mL for dehorning. For castration, TA was applied following extrusion of the testes and prior to severing the spermatic cords, by inserting the nozzle into the tunica vaginalis and delivering the product into the inguinal canal. For dehorning, it was applied directly onto the wounds immediately following the procedure. The method of application aimed to cover all injured tissue, including the spermatic cords which retract into the inguinal canal following the procedure.

### 2.4. Outcomes Measured

#### 2.4.1. Weight Gain (Experiment 1)

Weight gain was calculated for each calf using the difference of the pre-treatment weight collected on day 0 and the post-treatment weight collected on day 6.

#### 2.4.2. Behavioural Variables

##### Paddock Utilisation (Experiment 1)

GPS units were programmed using CatLog^TM^ software (Catnip Technologies Ltd., Anderson, CA, USA) to record a positional fix every 10 s using the Navstar global positioning system from 10:00:00 a.m. on day 0 for the entire experimental period (7 days). Location information was downloaded using the CatLog^TM^ software and exported into Microsoft Excel 2007. Only positional fixes recorded whilst all animals were in the paddock were included. Hence, all positional fixes before 08:00:00 a.m. on day 1 and after 11:59:59 p.m. on day 5 were disregarded. Positional fixes that were located outside the paddock boundary, which included a 40 m buffer to accommodate for possible large location errors associated with down antennas, short-fix intervals and sky obstructions [12], were removed. In addition, location fixes that were greater than 1 h apart or with a speed greater than 3.66 m/s [13] were removed. Paddock utilisation to determine 95% Minimum Convex Polygon (MCP) on a daily basis per animal was calculated in R 3.3.3 [14] using the ‘adehabitatHR’ package [15].

##### Lying Activity (Experiment 1)

The accelerometer loggers were pre-programmed using Onset HOBOware software (Onset Computer Corporation, Bourne, MA, USA) to record the g-force on the *x*-, *y*- and *z*-axes every 10 s from 10:00:00 a.m. on day 0. The loggers recorded until the memory was filled at 22:13:00 h on day 2. Following removal of the loggers, the data was downloaded using the Onset HOBOware software which converted the g-force readings into degrees of tilt. The data was then exported into Microsoft Excel 2007 and the degree of tilt on the *x*-axis was used to determine whether or not the calves were in a lying position at each 10-s reading. All data points prior to 12:00:00 p.m. on day 0 were removed as the last accelerometer unit was attached at 11:45:00 a.m. Tilt values >120° were interpreted as standing and tilt values ≤120° were interpreted as lying. These thresholds were based on values used in previous studies on dairy cows [16,17] and adjusted according to the orientation of the loggers on the legs of the animals. 

##### Behaviour (Experiment 2)

Calves remained in the holding yard for 6 h following treatment. During this time, calves were provided *ad libitum* access to water and lucerne hay. Six HD 1080p Sports Action Cam video cameras (Sony Australia Ltd., North Sydney, NSW, Australia), were attached at various points along the fence of the yard to capture video footage of the calves. Cameras were placed strategically to capture footage from all angles of the yard. This footage was later used to continuously record the frequency or duration of certain specified behaviours displayed by each animal in 5-min focal samples at 6 time points (40, 80, 120, 180, 240 and 360 min following treatment). The frequency and duration of behaviours were recorded by a single, trained observer using. The Observer^®^ XT 12 observational data software package (Noldus Information Technology, Wageningen, The Netherlands). The observer was blinded to treatment, although it was clear which calves were CONP calves due to the presence of intact horns. An ethogram was designed using The Observer^®^ XT software whereby behaviours were categorised as states or points (Table 1). State behaviours were quantified by duration (s) and point behaviours were quantified by frequency. The ethogram was derived from previous published studies on surgical castration and amputation dehorning [18,19,20,21].

### 2.5. Statistical Analysis

All data (Appendix A) were subjected to restricted maximum likelihood (REML) using Genstat^®^ 17th Edition statistical software (VSN International Ltd., Hemel Hempstead, UK). For weight gain, outliers within treatment groups were identified using the boxplot procedure of Genstat^®^. A linear mixed models procedure was used to analyse data on weight gain, paddock utilisation and observed state behaviours. A generalised linear mixed models (GLMM) procedure with a binomial distribution was used to analyse data on lying activity generated from accelerometer readings. A macro was used in Microsoft Excel 2007 to calculate the frequency of lying bouts and average duration of lying bouts. A GLMM procedure with a Poisson distribution was used to analyse data on frequency of lying bouts and a linear mixed models procedure was used to analyse data on average duration of lying bouts. A generalised linear mixed models (GLMM) procedure with a Poisson distribution was used to analyse data on observed point behaviours. For weight gain (experiment 1), the fixed effect of the model was treatment (CONP, CONN, BM, TA, BMTA). For paddock utilisation (experiment 1) the fixed effects of the model were treatment (CONP, CONN, BM, TA, BMTA) × day (1, 2, 3, 4, 5). For total lying activity (experiment 1), frequency of lying bouts and average duration of lying bouts, the fixed effects of the model were treatment (CONP, CONN, BM, TA, BMTA) × day (0, 1, 2) + BW (variate). For each observed behaviour (Table 1) (experiment 2), the fixed effects of the model were treatment (CONP, CONN, BM, TA, BMTA) × time-point (40, 80, 120, 180, 240, 360 min) + day (A, B, C) + BW (variate). The random effect for all models was calf ID. Insignificant terms were dropped from the models using a backwards elimination approach. Data on weight gain and observed behaviours is presented as predicted means. Data on lying activity is presented as the proportion of time calves spent lying. For all statistical calculations, *p* values ≤ 0.05 were considered statistically significant.

## 3. Results

### 3.1. Animals and Environment

For experiment 1, calves weighed 198.77 ± 36.39 kg at the beginning of the trial. Daily maximum temperature throughout this experiment was 21.4 °C, 21.1 °C, 20.7 °C, 23.8 °C, 19.7 °C, 19.9 °C and 23.6 °C for days 0, 1, 2, 3, 4, 5 and 6, respectively. Daily global solar exposure throughout this experiment was 13.5, 7.8, 15.0, 13.2, 14.9, 6.4 and 4.8 MJ/m^2^ for days 0, 1, 2, 3, 4, 5 and 6, respectively.

For experiment 2, calves weighed 206.88 ± 40.23 kg. Days A, B and C of experiment 2 correspond with days 3, 4 and 5 of experiment 1.

### 3.2. Weight Gain (Experiment 1)

Ten data points were excluded, as 3 (1 × CONN, 1 × TA and 1 × BMTA) were missing upon the second weighing and 7 (1 × CONP, 2 × CONN, 1 × BM and 3 × BMTA) were identified as outliers within their treatment groups using the boxplot procedure of Genstat^®^. 

There was a significant effect of treatment on weight gain (*p* < 0.001). CONP and BMTA calves had significantly greater weight gain values than CONN calves. CONP calves also had significantly greater weight gain values than BM and TA calves (Table 2).

### 3.3. Behavioural Variables

#### 3.3.1. Paddock Utilisation (Experiment 1)

As part of the data ‘cleaning’ procedures, 8.4% of the total data points were removed; 16.5%, 4.1%, 4.6%, 11.9% and 3.7% of the data points were removed for treatment groups CONP, CONN, BM, TA and BMTA, respectively. There was no significant effect of treatment on paddock utilisation (*p* = 0.167). While there was a significant effect of day on paddock utilisation (*p* < 0.001), this is not presented nor discussed further due to acknowledged logging time and duration differences across days.

#### 3.3.2. Lying Activity (Experiment 1)

There was no significant effect of body weight on total lying activity (*p* = 0.724). There was a significant interaction between treatment and day (*p* < 0.001) on total lying activity. CONN calves spent the least proportion of time lying and BMTA calves spent the greatest proportion of time lying on all days. All other calves spent an intermediate proportion of time lying compared to CONN and BMTA calves on all days. The proportion of time spent lying increased from day 0 to day 1 for all calves and again from day 1 to day 2 for all calves except CONP calves (Table 3). 

There was no significant effect of BW on the frequency of lying bouts or the average duration of lying bouts (*p* = 0.743 and *p* = 0.079, respectively). There was no significant effect of treatment on the frequency of lying bouts or the average duration of lying bouts (*p* = 0.225 and *p* = 0.141, respectively). While there was a significant effect of day on the average frequency of lying bouts and the average duration of lying bouts (*p* < 0.001 and *p* < 0.001, respectively), this is not presented nor discussed further due to acknowledged logging time and duration differences across days.

#### 3.3.3. Individual Behaviours (Experiment 2)

There were 6 missing focal samples due to calves being unidentified in the video footage. Of these missing samples, there was one from time point 1 (1 × BMTA calf), one from time point 2 (1 × BMTA calf) and 4 from time point 6 (1 × CONP, 1 × BM and 2 × TA calves). Behaviours influenced by time only are neither presented nor discussed. As the behaviours ‘walk with a stiff gait,’ ‘walk with a limp,’ ‘stand statue’ and ‘lie abnormal’ occurred infrequently, it was decided to only analyse the behaviours ‘walk,’ ‘stand’ and ‘lie,’ instead of their modifiers (‘walk relaxed,’ ‘walk with a stiff gait,’ ‘walk with a limp,’ ‘stand relaxed,’ ‘stand statue,’ ‘lie normal’ and ‘lie abnormal’). The behaviours head pawing and kicking occurred too infrequently for statistical analysis. 

There was a significant effect of treatment × time on the frequency of ear flicks (*p* = 0.006) displayed by the calves. The frequency of ear flicks was significantly greater in TA calves than in CONP, CONN and BMTA calves at 120 min and significantly greater in BM calves than in TA calves at 240 min (Table 4). 

There was a significant effect of treatment on the frequency of head turns (*p* = 0.049) and tail flicks (*p* = 0.04) displayed by calves. CONP calves displayed significantly less head turns than TA calves. CONP and TA calves displayed significantly less tail flicks than CONN and BM calves (Table 4). There was a significant effect of treatment on the duration of time calves spent walking (*p* = 0.024), eating (*p* < 0.001) and drinking (*p* = 0.002). The duration of time spent walking was significantly less in CONP calves than in CONN and BMTA calves and significantly greater in BMTA calves than in BM and TA calves. The duration of time spent eating was significantly greater in CONP calves than in all other calves and significantly less in TA calves than in BMTA calves. The duration of time spent drinking was significantly greater in CONP calves than in BMTA calves (Table 5). Treatment did not have a significant effect on the duration or frequency of any other behaviours. 

There was a significant effect of day on the duration of time calves spent drinking (*p* < 0.001). Calves treated on day 1 spent a greater duration of time drinking compared to calves treated on days 2 or 3 (Table 6). There was a significant effect of day on the frequency of head shakes (*p* < 0.001), head turns (*p* < 0.001), ear flicks (*p* < 0.001), stamps (*p* = 0.022) and tail flicks (*p* < 0.001) displayed by calves. Calves treated on day 1 displayed more head shakes, head turns and ear flicks than those treated on days 2 and 3. Calves treated on days 1 and 2 exhibited more foot stamps on than those treated on day 3. The frequency of tail flicks decreased each day (Table 6). Day did not have a significant effect on the duration or frequency of any other behaviours.

## 4. Discussion

Practical issues associated with injectable anaesthetics and analgesics have prevented their widespread uptake by Australian beef producers. However, as ‘farmer applied’ pain relief products are now commercially available for use on calves undergoing surgical husbandry procedures, this study investigated the effects of TA and MEL, separately and in combination, on weight gain, lying activity and individual behaviours following concurrent castration and dehorning of *Bos indicus* weaner calves. Topical anaesthetic allows delivery of lignocaine and bupivacaine via absorption at the wound site and MEL is absorbed through the mucosa of the buccal cavity. There are few previous studies investigating the effects of surgical husbandry procedures and pain relief on welfare of *Bos indicus* cattle [22,23,24,25,26,27,28]. In our study, results of experiments 1 and 2 have not been directly compared due to differences in animal numbers, dehorning and castration technicians and experimental environments and timeframes. The findings show a combination of TA and MEL improved short-term weight gain and increased lying activity following castration and dehorning, suggesting this combination of treatments was likely effective in improving the welfare of *Bos indicus* weaner calves. There were also behavioural trends suggesting TA and MEL reduced pain following castration and dehorning of *Bos indicus* weaner calves.

Assessment of production parameters following invasive husbandry procedures in livestock is important and of relevance to producers seeking to optimise welfare and production [29]. Weight gain and various measures of stress and pain have been used to evaluate animal welfare following castration and dehorning in calves [30,31,32]. In farm animals, pain can reduce feeding behaviour and invoke stress responses and immune reactions that affect production parameters, including weight gain [33]. For example, increased nociceptor activity increases sympathetic tone and adrenal secretions, potentially inhibiting gastric control centres, causing decreased rumen motility [34]. A reduction in weight gain is expected to follow castration and dehorning [35], suggesting poor animal welfare and economic losses from such procedures [32]. In the current study, all calves, including CONP calves, appeared to lose weight over the 6 days following treatment. This result may have been partly due to differences in feed allocation and gut fill between days 0 and 6, as calves were weaned and kept in holding yards with access to feed and water one week before procedure and were then moved to a large paddock to feed on available pasture on days 1 to 6 following treatment. Weight loss following treatment was greatest in CONN calves and lowest in CONP calves. This result aligns with previous findings showing concurrent castration and dehorning to negatively impact average daily gain (ADG) [31,35]. Weight change of BMTA calves did not differ significantly from that of CONP calves, indicating that a combination of TA and MEL may provide superior pain relief than TA or MEL separately. This finding is consistent with literature recommending a combination of LA and NSAIDs to target both the acute nociceptive and inflammatory phases of the pain response [36,37]. The weight gain results in our study support previous research findings, where calves had greater ADG values for the first 13 days following concurrent castration and dehorning when administered pain relief in the form of sodium salicylate or a combination of sodium salicylate, xylazine, ketamine and butorphanol, compared to no analgesic treatment [31]. 

In Australia, beef cattle producers are generally paid a monetary value per kg BW or carcass weight (cwt). The results presented demonstrate that a combination of TA and MEL can be a cost-effective addition to routine practice, whilst improving animal welfare [31]. For example, the current price of beef is approximately $3.50/kg live-weight. In this trial, the administration of TA and BM cost approximately $5 per calf (using the retail price of the therapeutics). CONN calves lost 2.9 kg BW more than BMTA calves, equating to a loss of $10.15 in value, indicating that the price of providing pain relief was less than the gain in product value from its use.

In cattle, GPS technology has mainly been used to monitor grazing behaviour [38,39,40]. This study attempted to use GPS location to identify possible changes to calf behaviour in relation to paddock utilisation as a response to pain. In this case, paddock utilisation was measured through calculation of 95% MCPs. The MCP is a frequently used technique for home-range calculation, which identifies a restricted area within which an animal moves when performing its normal activities [41]. The MCP technique has mostly been used in wildlife habitat studies, with little use in livestock studies so far [42]. In our study, there was no effect of treatment on 95% MCP values, suggesting concurrent castration and dehorning may have had no impact on the ability for calves to access and utilise available pasture resources across their landscape in the days following the procedures. It is likely that paddock utilisation was similar between all animals because of a social influence of peer activity on individual calf behaviour [43]. Pain may have had an effect on other behavioural measures, such as speed of movement, distance travelled and distance to peers, as these variables have been used to evaluate welfare in other species. In sheep, GPS technology has been used to identify lambing behaviour [44]. A decrease in daily speed and hourly speed following lambing and an increase in distance to peers during lambing was identified [44]. Additionally, GPS technology has been used in sheep to show a positive linear relationship between faecal egg count and distance moved per time step, suggesting that an increase in parasite load may result in animals grazing for longer periods or travelling to water more frequently [45]. In dogs, GPS technology has been used to distinguish between healthy dogs and dogs with osteoporosis through differences in performance measures [46]. Velocity, acceleration and deceleration were all reduced in dogs with osteoporosis compared to healthy dogs [46]. In addition, an improvement to these performance measures was shown in dogs with osteoporosis when oral carprofen was administered [46]. These studies reinforce the potential for GPS technology to identify production and welfare improvements in animals. In our study, the total number of data points that were removed as part of the ‘cleaning’ procedures prior to analysis was 8.4%, suggesting the accuracy of the positional fixes may not have been high. An estimation of paddock utilisation may require less accuracy than measurements of fine-scale dynamics of movement [12]. Hence in our study we chose to use 95% MCP as a measure of paddock utilisation due to the likeliness that the GPS positional fixes were not highly accurate. Future studies should employ the use of suitable GPS units to accurately measure other variables, such as speed and distance travelled, to assess potential effects of pain and pain relief in cattle. 

Accelerometers have been used to record activity of calves following surgical castration [47], disbudding and dehorning [48,49] and concurrent castration and dehorning [50]. As an increase or decrease in lying activity is not a direct measure of pain, such observations should be interpreted with caution. Lying activity exhibits a significant degree of individual variability in cattle [34] and it is likely that inter-animal comparisons from what is normal in the absence of pain [34] before treatment, compared to after treatment, may be a more sensitive measurement than between-animal comparisons. However, as inter-animal comparisons from before to after treatment would have required an additional round of mustering in the current study, between-animal comparisons were used for practical reasons. Although the analysis failed to find a significant difference between CONN and CONP calves for lying activity, the overall results and trends for this outcome suggest that less lying may be indicative of greater discomfort or pain. This finding agrees with the results of previous studies using accelerometers or behavioural observations to monitor lying activity of calves undergoing castration or dehorning [19,34,47,48,49]. Surgically castrated calves have previously been shown to spend more time standing following the procedure, compared to pre-operatively, as measured using accelerometers [47]. Similarly, accelerometer measurements have shown that dehorning in calves reduces lying activity, which is less significant or not apparent when meloxicam has been administered [34,49]. In future research, it would be beneficial to further classify standing activity as ‘immobile’ or ‘mobile/walking,’ as this could highlight potential differences between treatment groups that were unknown in the current study. However, this would require a higher sampling rate, subsequently reducing the memory storage of recording devices and limiting the time period for data collection. The increase in lying activity seen in all calves from day 0 to day 1 can be explained by the restriction of calves to the holding yards and laneway on day 0 and the increased sampling time on day 1. The calves may have been less inclined to lie down in this environment compared to a paddock environment, as ground cover in the laneway mainly consisted of dirt. In addition, there were humans present near the laneway during daytime hours on day 0, potentially deterring the calves from resting. As the increase in lying activity from day 1 to day 2 was only seen in castrated and dehorned calves, it may indicate a reduction in discomfort or pain over time. 

Observation of individual behaviours has previously been used to measure pain following castration [19,21], dehorning [18,20] and concurrent castration and dehorning [51]. These studies have also used the analysis of individual behaviours to evaluate the efficacy of local anaesthesia and analgesia for these procedures [18,19,20,51]. In experiment 2, calves that had been castrated and dehorned spent a significantly greater duration of time walking and a significantly less duration of time eating compared to CONP calves. Excessive locomotion, as demonstrated in this study through increased time spent walking, is recognised as a pain-related behaviour [24,33]. It is unclear why BMTA calves spent more time walking compared to BM and TA calves. Pain-related behaviour and behavioural responses to certain procedures is variable between individual animals [22,34,52] and may explain this finding. Pain in animals has the potential to reduce eating behaviour in animals [33]. A previous study showed that control calves spent more time eating than castrated and dehorned calves and that a combination of lignocaine and flunixin meglumine, increased the amount of time spent eating [51]. In experiment 2 of the current study, CONP calves spent more time eating than all other calves and there was a trend for BMTA calves to spend more time eating than CONN calves, suggesting a reduction in pain with a combination of TA and MEL. In experiment 2 of the current study, calves that had been castrated and dehorned tended to display a greater frequency of tail flicks than CONP calves. An increased frequency of tail flicks has previously been observed for these procedures performed both singularly [20,53] and in combination [51] and is suggested to be due to irritation or pain [20,51,53]. TA calves did not differ from CONP calves in their display of tail flicks and there was a trend for BMTA calves to display less tail flicks in comparison to CONN and BM calves. This finding suggests that TA may have reduced pain. There was a significant interaction between treatment and time on the frequency of ear flicks and a significant effect of treatment on the duration of time spent drinking and the frequency of head turns, although there was no clear trend in this data. Again, potential variation between individual animals in regard to expression of these behaviours may have influenced these results. With ear flicks, it is possible that the procedures of ear tagging and notching, with the latter procedure known to cause substantial pain [54] may have confounded these results. In addition, the display of certain behaviours seemed to be influenced by other factors independent of pain. This is evident in the significant effect of day on some behaviours, such as the duration of time that calves spent drinking and the frequency of head shakes, head turns, stamps, ear flicks and tail flicks. It was noted that more crows and flies were present in the vicinity of the calves treated on day 1 compared to those treated on days 2 and 3. Differences in weather conditions are likely to explain this observation, with day 1 being hotter and less overcast than days 2 and 3. As discussed above, although there were some behaviours that appeared to be associated with pain, as demonstrated through a difference between CONN and CONP calves, overall, there was limited expression of pain-related behaviours displayed by the calves in this study. It has been suggested that the age and breed of animals influences their behavioural demonstration of pain and thus affects observations on methods for relief of pain [55]. Dairy calves appear to display more prominent responses to painful procedures and pain relief interventions compared to beef cattle, particularly when the beef calves are from environments where predation occurs commonly and animals quickly learn to minimise their demonstrations of pain [55]. The calves used in this study are likely to have had a strong tendency to hide their expression of pain. The majority of the previous literature on the behavioural response to castration and dehorning of cattle has used younger dairy calves [56,57], with minimal research having been conducted using older *Bos Indicus* beef calves [24]. In addition, there is very little research that has examined the behavioural response to castration and dehorning of calves, when performed concurrently [51]. Therefore, the results of this study provide novel information on the behaviour of weaned *Bos Indicus* calves following concurrent castration and dehorning.

This study may be the first documented examination of the effects of TA and MEL following concurrent castration and dehorning of weaner calves. It should be considered that the stressful experiences of handling, weaning and concurrently performed surgical procedures may have had an effect on the results of this study, especially as these calves had none or very little prior interaction with humans and handling facilities. The importance of conducting studies that closely represent current industry practice and the possible changes to it has previously been acknowledged [22] and is emphasised in the current study. The difficulty in obtaining consistent results across all measures of pain and for all treatments is a common issue in studies on animal pain and may be especially apparent in studies on *Bos indicus* cattle which are usually unaccustomed to humans and handling [22,23]. 

## 5. Conclusions

In experiment 1, a significant improvement in weight gain was seen following castration and dehorning when a combination of TA and MEL had been administered at the time of marking, resulting in no difference between CONP and BMTA calves. This experiment also found a combination of TA and MEL increased lying activity in the first few days following treatment, suggesting a reduction in pain. In experiment 2, there were trends for TA and the combination of TA and MEL to reduce pain-related behaviours during a 6-h period following castration and dehorning that warrant further investigation. Overall, an improvement in weight gain, an increase in lying activity and behavioural trends indicative of efficacy demonstrate the potential for TA and MEL to improve welfare and production following castration and dehorning of beef calves. This is an important finding for large, extensive tropical beef production systems that are seeking practical options for improving animal welfare.

## Figures and Tables

**Table 1 animals-08-00035-t001:** Ethogram developed for behavioural observations conducted on calves following treatment.

Behaviour	Description
**States ^1^**	
Walk	Walking forwards or backwards in any style at any pace.
Stand	Standing in any style.
Lie	Lying down completely on the ground in any style.
Head down	Holding head below brisket.
Eat	Ingesting lucerne hay.
Drink	Ingesting water.
**Points ^2^**	
Head shake	Rapid shaking of the head around a rostral to caudal axis.
Head turn	Rapid turning of the head to either side of the body.
Head paw	Lifting of hind leg and contacting the head.
Kick	Kicking backward or towards the belly with a hind limb.
Stamp	Lifting front or hind foot and forcefully placing it on the ground.
Ear flick	Rapid movement of one or both ears.
Tail flick	Sideways movement of the tail from vertical to return to vertical.

^1^ States are behaviours with measurable duration and are quantified by duration of time (s). ^2^ Points are behaviours without measurable duration and are quantified by frequency.

**Table 2 animals-08-00035-t002:** Mean weight gain of calves in each treatment group over 6 days.

Treatment	Mean Weight Gain (kg) ± s.e.m.
CONP	−3.69 ^a^ ± 0.77
(*n* = 50)	
CONN	−8.30 ^c^ ± 0.77
(*n* = 50)	
BM	−6.62 ^bc^ ± 0.76
(*n* = 50)	
TA	−6.59 ^bc^ ± 0.76
(*n* = 50)	
BMTA	−5.40 ^ab^ ± 0.79
(*n* = 50)	

CONP = no castration and dehorning/positive control; CONN = castration and dehorning/negative control; BM = castration and dehorning with pre-operative buccal meloxicam; TA = castration and dehorning with intra-operative topical anaesthetic; and BMTA = castration and dehorning with pre-operative buccal meloxicam and intra-operative topical anaesthetic. ^a^, ^b^, ^c^ Values with different superscripts differ significantly at *p* ≤ 0.05. Descriptive statistics are based on predicted means (±s.e.m.). A significant effect was found (*p* < 0.001).

**Table 3 animals-08-00035-t003:** Proportion of time spent lying by calves in each treatment group on days 0, 1 and 2.

Day	Proportion of Time Spent Lying Down (%)				
	CONP	CONN	BM	TA	BMTA
	(*n* = 10)	(*n* = 10)	(*n* = 10)	(*n* = 10)	(*n* = 10)
0	30.09 ^Aab^ ± 0.37	16.55 ^Aa^ ± 0.46	39.11 ^Aab^ ± 0.35	29.53 ^Aab^ ± 0.37	50.46 ^Ab^ ± 0.26
1	50.57 ^Bab^ ± 0.26	24.84 ^Ba^ ± 0.42	44.37 ^Bab^ ± 0.32	41.63 ^Bab^ ± 0.30	66.81 ^Bb^ ± 0.17
2	49.19 ^Bab^ ± 0.27	27.64 ^Ca^ ± 0.40	45.81 ^Cab^ ± 0.31	43.58 ^Cab^ ± 0.29	67.80 ^Cb^ ± 0.17

CONP = no castration and dehorning/positive control; CONN = castration and dehorning/negative control; BM = castration and dehorning with pre-operative buccal meloxicam; TA = castration and dehorning with intra-operative TA; and BMTA = castration and dehorning with pre-operative buccal meloxicam and intra-operative topical anaesthetic. ^a^, ^b^ Values within a row with different superscripts differ significantly at *p* ≤ 0.05. ^A^, ^B^, ^C^ Values within a column with different superscripts differ significantly at *p* ≤ 0.05. Descriptive statistics are based on predicted means (±s.e.m.). A significant effect was found (*p* < 0.001).

**Table 4 animals-08-00035-t004:** Mean frequency of ear flicks, head turns and tail flicks displayed by calves in each treatment group within a 5-min focal sample at each time-point.

Behaviour	Effect and *p*-Value	Time (min)	CONP	CONN	BM	TA	BMTA
			(*n* = 12)	(*n* = 11)	(*n* = 11)	(*n* = 11)	(*n* = 12)
Ear flicks	Treatment × Time (*p* = 0.006)	40	0.53 ^Aba^ ± 0.31	1.84 ^Aa^ ± 0.71	0.66 ^Aa^ ± 0.35	1.59 ^Ba^ ± 0.61	0.50 ^Aba^ ± 0.30
		80	0.20 ^Aa^ ± 0.18	0.80 ^Aa^ ± 0.42	0.86 ^Aba^ ± 0.41	0.25 ^Aa^ ± 0.20	0.14 ^Aa^ ± 0.15
		120	0.27 ^Aa^ ± 0.21	0.56 ^Aa^ ± 0.34	0.72 ^ABab^ ± 0.37	3.24 ^Bb^ ± 1.05	0.48 ^Aba^ ± 0.29
		180	0.53 ^Aba^ ± 0.31	0.80 ^Aa^ ± 0.42	1.78 ^Aba^ ± 0.66	0.89 ^Aba^ ± 0.41	0.41 ^Aba^ ± 0.27
		240	0.47 ^ABab^ ± 0.28	1.36 ^Aab^ ± 0.58	2.57 ^ABb^ ± 0.87	0.38 ^Aa^ ± 0.25	0.55 ^ABab^ ± 1.22
		360	1.12 ^Ba^ ± 0.50	0.72 ^Aa^ ± 0.39	3.31 ^Ba^ ± 1.09	2.14 ^Ba^ ± 0.96	0.68 ^Ba^ ± 0.36
Head turns	Treatment (*p* = 0.049)		0.52 ^a^ ± 0.15	0.97 ^ab^ ± 0.24	1.04 ^ab^ ± 0.26	1.42 ^b^ ± 0.33	0.57 ^a^ ± 0.28
Tail flicks	Treatment (*p* = 0.04)		2.95 ^a^ ± 0.92	7.73 ^c^ ± 2.16	9.65 ^c^ ± 2.65	3.95 ^ab^ ± 1.21	6.13 ^bc^ ± 1.67

CONP = no castration and dehorning/positive control; CONN = castration and dehorning/negative control; BM = castration and dehorning with pre-operative buccal meloxicam; TA = castration and dehorning with intra-operative topical anaesthetic; and BMTA = castration and dehorning with pre-operative buccal meloxicam and intra-operative topical anaesthetic. ^a^, ^b^, ^c^ Values within a row with different superscripts differ significantly at *p* ≤ 0.05. ^A^, ^B^ Values within a column with different superscripts differ significantly at *p* ≤ 0.05. Descriptive statistics are based on predicted means (±s.e.m.).

**Table 5 animals-08-00035-t005:** Mean duration of time (s) spent walking, eating and drinking by calves in each treatment group within a 5-min focal sample.

Behaviour	Effect and *p*-Value	CONP	CONN	BM	TA	BMTA
		(*n* = 12)	(*n* = 11)	(*n* = 11)	(*n* = 11)	(*n* = 12)
Walking	Treatment (*p* = 0.024)	23.82 ^a^ ± 6.62	47.09 ^bc^ ± 6.90	36.89 ^ab^ ± 6.92	32.78 ^ab^ ± 6.93	53.45 ^c^ ± 6.64
Eating	Treatment (*p* < 0.001)	127.64 ^a^ ± 14.00	33.01 ^bc^ ± 14.55	48.73 ^bc^ ±14.63	18.98 ^c^ ±14.71	67.88 ^b^ ± 14.08
Drinking	Treatment (*p* = 0.002)	9.43 ^a^ ± 1.86	5.30 ^ab^ ± 1.92	6.39 ^ab^ ± 1.95	2.65 ^ab^ ± 1.96	1.20 ^b^ ± 1.87

CONP = no castration and dehorning/positive control; CONN = castration and dehorning/negative control; BM = castration and dehorning with pre-operative buccal meloxicam; TA = castration and dehorning with intra-operative topical anaesthetic; and BMTA = castration and dehorning with pre-operative buccal meloxicam and intra-operative topical anaesthetic. ^a^, ^b^, ^c^ Values within a row with different superscripts differ significantly at *p* ≤ 0.05. Descriptive statistics are based on predicted means (±s.e.m.).

**Table 6 animals-08-00035-t006:** Mean duration of time (s) spent drinking and mean frequency of head shakes, head turns, stamps, ear flicks and tail flicks displayed by calves (*n* = 57) on each day within a 5-min focal sample.

Behaviour	*p*-Value	Outcome	Day A	Day B	Day C
			(*n* = 19)	(*n* = 19)	(*n* = 19)
Drinking	*p* < 0.001	Duration of time (s)	12.43 ^a^ ± 1.56	2.43 ^b^ ± 1.45	0.12 ^b^ ± 1.44
Head shakes	*p* < 0.001	Frequency	1.44 ^a^ ± 10.61	0.46 ^b^ ± 3.37	0.28 ^b^ ± 2.10
Head turns	*p* < 0.001	Frequency	1.62 ^a^ ± 0.30	0.60 ^b^ ± 0.13	0.61 ^b^ ± 0.13
Stamps	*p* = 0.022	Frequency	0.21 ^a^ ± 0.07	0.17 ^a^ ± 0.06	0.06 ^b^ ± 0.02
Ear flicks	*p* < 0.001	Frequency	1.57 ^a^ ± 0.34	0.52 ^b^ ± 0.13	0.51 ^b^ ± 0.13
Tail flicks	*p* < 0.001	Frequency	11.45 ^a^ ± 2.47	5.39 ^b^ ± 1.18	2.79 ^c^ ± 0.68

^a^, ^b^ Values within a row with different superscripts differ significantly at *p* ≤ 0.05. Descriptive statistics are based on predicted means (±s.e.m.). Body weight did not have a significant effect on the duration or frequency of any behaviours.

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
