# Peer review of "Effects of Topical Anaesthetic and Buccal Meloxicam Treatments on Concurrent Castration and Dehorning of Beef Calves"

_animals, 2018, doi:10.3390/ani8030035_

Round 1

Reviewer 1 Report

I enjoyed reading your paper, which was well designed and well presented. I have only a few minor suggestions for improvement - attached.

Author Response

Line 84 - The sentence summarising the findings at the end of the introduction has been removed.

Line 365 – Change has been made.

Line 372 – In the paper by Musk et al. (2017), there was no significant effect of the castration procedure on change in weight. There are also numerous other previous studies that have found no effect of castration or dehorning. There are also numerous studies that have found a negative effect of the procedures on change in body weight as compared to control animals (as in our study). The authors have chosen to focus on concurrent castration and dehorning in regards to this point to narrow the field of comparison. In light of this, the authors have now deleted the sentences at the end of this paragraph that refer only to castration or dehorning.

Line 413 – This is a good point made by the reviewer. However, in the studies by Musk et al. (2017) and Laurence et al. (2017), pedometers were used to measure steps taken and resting activity. The authors argue that resting activity and lying activity differ, assuming that resting behaviour is categorised as a pause in activity/steps taken, and not by a posture (standing/lying). This emphasises the need to look at walking, standing and lying separately, as noted in our paper.

Line 421 – This sentence has been changed to clarify what we are referring to.

Reviewer 2 Report

Overall, these studies looked at the effect of anaesthetic and/or analgesic treatment on different variables that may indicate production and welfare improvement following concurrent dehorning and surgical castration. The researchers suggest that buccal meloxicam and topical anaesthetic gel combination can result in more weight gains and less pain responses than their separate administration, and a negative control following the procedures. However, the combination treatment did not produce strong and significant differences in the variables measured except a few, when compared with the positive control and other treatments. On the other hand, this combination group showed one or two pain related behaviours, e.g. more time walking, during the first six hours post-operatively. Authors can just indicate that the BMTA treatment has the potential to improve production and welfare of the calves after concurrent dehorning and castration procedures.  

Introduction

Lines 53- 54: check grammar

Lines 65-68: long, complex sentence.

Line 68: “injectable.”

Line 74: delete comma.

Line 76: insert “drug” after anti-inflammatory.

Line 77: did you mean it is registered “for use during” surgical castration…..?

Lines 79-81: you did not precisely mention the time of administration of TA and MEL to compare with pre-surgical injections.

A small paragraph on literature relevant to TA and MEL, separately and/or combination, in farm animals might be useful.

Line82: ‘Separately,” is a better word than “alone”; also, if used elsewhere in the text.

Lines 84-86: hypothesis fits better than findings here. Again, you can make a hypothesis if you review the literature on use of TA and MEL (buccal), separately and/or in combination.

Materials and methods

Lines 101 -102: TA administered post-operatively. In lines 167-168 TA was applied following skin incision, extrusion of the testes and prior to severing the cord….. Isn’t it an intra-operative application?

Line 149:  operator variability on the procedure and the resultant pain intensity? Did you have the same operators for both experiments?

Line220: all data “were”

Discussion

Line 342: practical issues “associated” with “injectable”

Line 348 -349: grammar check. Delete “treated” before calves as you already said “combination of treatments.”

Lines 354-357: long and complex sentence. Cut short.

Line 360: delete “result” from such procedures.

Lines 361-363: one week before procedure. When did you allocate calves to treatment groups? Did you look for significant changes in weight loss when calves were kept in holding yards and before day 0 i.e. start of procedures?

Line 365: add “following the procedures.”

Line: 409-410: you would have measured the fine-scale dynamics of movement, at least for a subset in each treatment group. Although calves with pain may follow their peers (with less or more pain, or no pain) due to social influence, fine-scale measurements such as speed of movement, distance travelled etc. would give some useful data to find out treatment differences! Discuss “why” you could not measure fine details.

Lines 408- 409: to make it clearer to the reader - better to explain briefly (one or two lines) how MCP relates to paddock utilisation than instant references, and insert a figure (polygon) to reflect your MCP results!

It would be interesting to measure the day-to-day GPS activity to gauze betterment over 6 days!

Lines 429-430:  BMTA calves had limited grazing activity compared to CONP calves but the line 367 says no significant difference in weight gains between BMTA and CONP groups - conflicting statements! Do you propose any mechanisms for weight gains other than grazing in treated calves?

Line 457: Check the verb “lower.”

Lines 459 -461: how can you explain the treatment effects (in BMTA group) in relation to individual behaviours that are variable?

Line 466: how could BMTA calves spend more time walking and eating? Did you mean they ate while walking?

Lines 467 – 469: it is not appropriate to relate the results re eating from the second experiment to the weight gains observed in the first experiment. Because you observed weight gains over 6 days post-procedures in the first experiment unlike the second experiment where you collected data on eating and other behaviours over 6 hours and/or 3 days from treatment groups.

A few lines on the mechanism of action and pharmacokinetics of the two drugs would be useful in the discussion. A brief mention of the intensity/severity of pain due to two concurrent procedures and pre- and post-procedural stress on weaner calves that may influence the treatment effects will be useful. As these results are from subsets of treatment population, the authors should state this in the conclusion to justify their suggestion on future studies with large number of animals.

Author Response

Lines 53- 54: Grammatical changes have been made and sentences have been restructured.

Lines 65-68: Sentence has been shortened.

Line 68: Change has been made.

Line 74: Change has been made.

Line 76: Change has been made.

Line 77: Yes – sentence has now been changed.

Lines 79-81: Time of administration of TA and MEL has been mentioned and compared with pre-surgical injections.

An expansion of literature relevant to TA and MEL in beef calves has been added to the paragraph on these products to justify why the current study aimed to further investigate these pain relief interventions in cattle.

Line 149: Operators did vary between experiments 1 and 2. This information has now been added to the materials and methods and has been acknowledged in the concluding paragraph of the discussion.

Line 220: Change has been made.

Line 342: Change has been made.

Line 348-349: Change has been made.

Line 354-357: Sentence has been simplified and shortened.

Line 360: Change has been made.

Lines 361-363: The number “1” has been changed to the word “one”. Calves were allocated to treatment in the order that they came through the race. This information has now been added in section 2.1. We didn’t look for significant changes in weight loss when the calves were kept in holding yards. As we had 250 calves randomly allocated to treatment groups, we didn’t think it was necessary to look at weight before the start of procedures.

Line 365: Change has been made.

Line 409-410: We thought measurements of fine scale dynamics would be inaccurate based on the amount of data we removed during the data cleaning process. The reason why we didn’t measure fine scale dynamics has now been expanded on in this paragraph.

Lines 408-409: Information on MCP and the relationship between paddock utilisation has been added. A figure of the polygon has not been added as the person responsible for analysis of this dataset has been unavailable during the time frame for resubmission. A figure can be obtained and sent through at a later date, if still required.

The authors chose to not present or discuss an effect of day on MCP, as the results for this were looked at and there was no explainable trend in the data in relation to pain or betterment over the days. The trend appeared to be random and possibly due to variations in environmental conditions over those days of which the authors have no record of.

Lines 429-430: The authors have decided to remove the lines discussing the trend for BMTA calves to spend more time lying than CONP calves as it is complex and only based on a trend.

Lines 459-461: The authors are unclear of what this question is asking. There are previous studies that have acknowledged the potential effects of individual variability on treatment effects. It is unlikely that the combination of treatments TA and MEL would have resulted in more pain than treatment with TA or MEL, separately.

Line 466: Behaviours were not looked at in regards to overall time budget. Instead, individual behaviours were compared between treatments and timepoints. It is possible that BMTA calves spent more time walking and more time eating and these behaviours didn’t necessarily have to occur at the same time.

Lines 467-469: All comparisons between experiments 1 and 2 have now been deleted.

There is no published information on the pharmacokinetics of Tri-Solfen and Buccalgesic. All other comments have been addressed through additions to the opening or concluding paragraphs of the discussion.

Reviewer 3 Report

I found this manuscript to provide information that is really valuable to the beef production industry, and commend you on the thoroughness with which you have approached this study.

I have one major comment (but easily addressed), and a couple of more minor comments.

I found the results section a bit confusing, particularly when in section 3.3.3 you started discussing effect of 'Day' - after I went back over the study design I realized that you were talking about 'Treatment Day' in experiment 2, as opposed to 'Day' following procedure in Experiment 1; so this needs to be clarified. How about splitting the results section into two, Experiment 1 and Experiment 2 (and in passing the 'animals and environment' data for Experiment 2 also needs to be added), and re-labeling the treatment days of Experiment 2 into A, B and C (or as Roman numerals) to differentiate them from the numbered 'Days' in Experiment 1.

Line 129: please give brand and manufacturer of the adhesive bandage and gaffer tape

Line 145: please give brand and manufacturer of the spray paint

Author Response

Each outcome measured is now specified as ‘Experiment 1’ or ‘Experiment 2’ in brackets. Days 1, 2 and 3 of Experiment 2 have been changed to Days A, B and C. Results on 'animals and environment' for experiment 2 has been added.

Line 129: Brands and manufacturers of the adhesive bandage and cloth tape have been added.

Line 145: Brand and manufacturer of the spray paint has been added.